



# Tropospheric NO₂ vertical profiles over South Korea and their relation to oxidant chemistry: Implications for geostationary satellite retrievals and the observation of NO₂ diurnal variation from space

Laura Hyesung Yang[1], Daniel J. Jacob[1,2], Nadia K. Colombi[2], Shixian Zhai[1], Kelvin H. Bates[1,3], Viral Shah[4],
Ellie Beaudry[1], Robert M. Yantosca[1], Haipeng Lin[1], Jared F. Brewer[5], Heesung Chong[6], Katherine R. Travis[7],
James H. Crawford[7], Lok Lamsal[8,9], Ja-Ho Koo[10], Jhoon Kim[10]

[1] Harvard University, John A. Paulson School of Engineering and Applied Sciences, Cambridge, MA 02138, USA
[2] Harvard University, Department of Earth and Planetary Sciences, Cambridge, MA 01238, USA
[3] University of California Davis, Department of Environmental Toxicology, Davis CA 95616, USA
[4] Global Modeling and Assimilation Office, NASA Goddard Space Flight Center, Greenbelt, MD 20771, USA, and Science Systems and Applications, Inc., Lanham, MD 20706, USA
[5] University of Minnesota, Department of Soil, Water and Climate, St. Paul, Minnesota, USA
[6] Harvard-Smithsonian Center for Astrophysics, Cambridge, Massachusetts 02138, USA
[7] NASA Langley Research Center, Hampton, VA 23666, USA
[8] Atmospheric Chemistry and Dynamics Laboratory, NASA Goddard Space Flight Center, Greenbelt, MD 20771, USA
[9] University of Maryland Baltimore County, Baltimore, MD 21250, USA
[10] Yonsei University, Department of Atmospheric Sciences, Seoul, South Korea

*Correspondence to*: Laura Hyesung Yang (laurayang@g.harvard.edu)

**Abstract.** Tropospheric nitrogen dioxide (NO₂) is of central importance for air quality, climate forcing, and nitrogen deposition to ecosystems. The Geostationary Environment Monitoring Spectrometer (GEMS) is now providing high-density NO₂ satellite data including diurnal variation over East Asia. The NO₂ retrieval requires independent vertical profile information from a chemical transport model (CTM) to compute the air mass factor (AMF) that relates the NO₂ column along the line of sight to the NO₂ vertical column. Here, we use aircraft observations from the Korea-United States Air Quality (KORUS-AQ) campaign over the Seoul Metropolitan Area (SMA) and around the Korean peninsula to better understand the factors controlling the NO₂ vertical profile, its diurnal variation, the implications for the AMF, and the ability of the GEOS-Chem CTM to compute the AMF and its variability. Proper representation of oxidant chemistry is critical for the CTM simulation of NO₂ vertical profiles and is achieved in GEOS-Chem through new model developments including aerosol nitrate photolysis, reduced uptake of hydroperoxy (HO₂) radicals by aerosols, and accounting for atmospheric oxidation of volatile chemical products (VCPs). We find that the tropospheric NO₂ columns measured from space are mainly contributed by the planetary boundary layer (PBL) below 2 km altitude, reflecting the highly polluted conditions. Repeated measurements of NO₂ vertical profiles over SMA at different times of day show that diurnal change in mixing depth affecting the NO₂ vertical profile induces a diurnal variation in AMF of comparable magnitude to the diurnal variation in the NO₂ column. GEOS-Chem captures this diurnal variation in AMF and more generally the variability in the AMF for the KORUS-AQ NO₂ vertical profiles (2.7% mean bias, 7.6% precision), with some outliers in the morning due to non-systematic errors in the timing of mixed layer growth.



## 1. Introduction


Nitrogen oxide radicals ($NO_x \equiv NO + NO_2$) are emitted by fuel combustion and from natural sources such as lightning, wildfires, and soils. They play a critical role in driving atmospheric oxidant chemistry with implications for air quality and health, climate forcing, and nitrogen deposition to ecosystems. Satellite-based retrievals of tropospheric $NO_2$ columns have been used extensively to investigate $NO_x$ emissions and their

trends (Richter et al., 2005; Stavrakou et al., 2008) and $NO_x$ atmospheric lifetime (de Foy et al., 2015; Laughner and Cohen, 2019). The retrievals require local $NO_2$ vertical profile information to account for atmospheric scattering, but there can be large uncertainties in these profiles (Travis et al., 2016). The Geostationary Environment Monitoring Spectrometer (GEMS) launched in February 2020 is now starting to provide continuous $NO_2$ column observations over East Asia including diurnal variation (J. Kim et al., 2020;

J. Park et al., 2022). Here we use observations from the Korea-United States Air Quality (KORUS-AQ) aircraft campaign together with GEOS-Chem chemical transport model (CTM) simulations to better understand the factors controlling $NO_2$ vertical profiles over East Asia, their relations to the broader photochemical environment, and the implications for GEMS retrievals.

Tropospheric $NO_2$ has been measured from polar sun-synchronous low-earth orbiting (LEO) satellite

instruments since 1995 with the Global Ozone Monitoring Experiment (GOME) (1995-2003; Burrows and Chance, 1993; Martin et al., 2002) continued by the Scanning Imaging Spectrometer for Atmospheric Chartography (SCIAMACHY) (2002-2012; Bovensmann et al., 1999), GOME-2 (2006-; Callies et al., 2000), the Ozone Monitoring Instrument (OMI) (2004-; Levelt et al., 2018), the Ozone Mapping Profiler Suite (OMPS) (2011-; Flynn et al., 2014), the Tropospheric Monitoring Instrument (TROPOMI) (2017-; Veefkind

et al., 2012), and the Environmental Trace Gases Monitoring Instrument (EMI) (2018-; Zhang et al., 2018). GEMS is the first geostationary instrument. The retrievals fit the backscattered solar spectra in the 400-470 nm wavelength range to obtain a $NO_2$ slant column density (SCD) from which the stratospheric portion is removed (Bucsela et al., 2013; Boersma et al., 2018). The tropospheric SCD is then converted to a vertical column density (VCD) with an air mass factor (AMF) for the scattering atmosphere that depends on the shape

of the local $NO_2$ vertical profile (Palmer et al., 2001; Eskes and Boersma, 2003). The vertical profile varies locally as a function of emissions, chemistry, and meteorology and must therefore be provided by CTM. Model errors in that vertical profile can be a major contributor to the VCD error budget (Martin et al., 2002; J. Lin et al., 2012; Boersma et al., 2018).



The LEO instruments observed $NO_2$ once a day at a particular local time of day, but tropospheric $NO_2$ VCDs

vary with the time of day as driven by emissions, chemistry, and mixing depth. Previous studies obtained

some sparse information on this diurnal variation by using the difference in overpass time between two LEO

satellites. Boersma et al. (2008) used SCIAMACHY and OMI and found higher tropospheric $NO_2$ VCDs in

mid-morning than in the early afternoon over polluted regions, mainly driven by photochemical loss over the

course of the day. J. Lin et al. (2010) used the diurnal variation of $NO_2$ as observed by GOME-2 and OMI

for a top-down estimate of $NO_x$ emissions over China including diurnal variation. Penn and Holloway (2020)

found that $NO_2$ column ratios between morning and afternoon are lower than surface $NO_2$ concentration

ratios, as would be expected from deeper vertical mixing in the afternoon, but with significant heterogeneity

in the ratio at the urban scale. These diurnal variations inferred from LEO satellite data seem however

inconsistent with observations from ground-based Pandora spectrometers over South Korea, which show flat

tropospheric $NO_2$ VCDs between mid-morning and afternoon (Chong et al., 2018; Crawford et al., 2021).

The hourly data from GEMS should enable a better understanding of this issue and an opportunity to better

account for the diurnal variation of $NO_x$ emissions and chemical lifetime.

The role of oxidant chemistry in controlling $NO_2$ and its vertical profile is of particular interest. This

chemistry involves ozone ($O_3$), hydrogen oxide radicals ($HO_x \equiv OH$ + peroxy radicals), and volatile organic

compounds (VOCs), which control the $NO/NO_2$ partitioning and the atmospheric lifetime of $NO_x$. CTM

errors in describing oxidant chemistry can result in large errors in $NO_2$ profiles (Travis et al., 2016; Silvern

et al. 2018). R. Park et al. (2021) found that the $O_3$ concentrations observed in KORUS-AQ were severely

underestimated by CTMs and this would affect the simulation of the $NO/NO_2$ ratio.

The KORUS-AQ campaign deployed the NASA DC-8 aircraft over and around the Korean peninsula in May-

June 2016 with vertical profiling at up to 7 km altitude (Crawford et al., 2021). The aircraft carried a large

payload of instrumentation for oxidant chemistry. Previous relevant analyses of the KORUS-AQ data focused

on comparisons to an ensemble of CTMs (R. Park et al., 2021), investigations of the factors controlling CO

and $O_3$ (Gaubert et al., 2020; Schroeder et al., 2020), heterogenous chemistry (Heim et al., 2020; Brune et

al., 2022), VOCs reactivity (Simpson et al., 2020), aerosol nitrate photolysis (Romer et al., 2018), and aerosol

composition (Nault et al., 2018; H. Kim et al., 2018; Jordan et al., 2020), along with a number of GEOS-

Chem-specific studies referenced in Section 2.

Here we conduct simulations of the KORUS-AQ observations using GEOS-Chem at $0.25° \times$

$0.3125°$ resolution with recent chemical updates including aerosol nitrate ($NO_3^-$) photolysis (Shah et al.,

2022), coarse particulate matter uptake of nitric acid ($HNO_3$) (Zhai et al., 2022a), emissions of volatile

chemical products (VCPs) (Bates et al., 2022) and revision of $HO_2$ heterogeneous uptake by aerosols. As we





will see, these updates enable a much-improved representation of oxidant chemistry relative to previous GEOS-Chem versions. We start from an evaluation of the model with observed vertical profiles of $O_3$ and $HO_x$ species which leads us to evaluation with observations of $NO_2$, NO, and $NO/NO_2$ ratios. We go on to examine the diurnal variation of the $NO_2$ AMF as defined by observed and GEOS-Chem vertical profiles,

and from there quantify the errors associated with using GEOS-Chem to compute the AMFs for GEMS retrievals.

## 2. Methods

### 2.1. KORUS-AQ observations

KORUS-AQ included 20 research flights with the NASA DC-8 aircraft, all based out of Osan Air Base and

covering South Korea and the adjacent ocean with a particular focus on the Seoul Metropolitan Area (SMA) (Crawford et al., 2021). All flights were in the 08 – 16 local time (LT) window. We exclude observations made near Daesan (36.4 – 37.15°N, 126 – 126.88°E) that focused on sampling power plant plumes. We use a 60-second merged dataset of the aircraft observations (available at https://www-air.larc.nasa.gov/). We sample the GEOS-Chem model along the flight tracks at the time of the observations. Comparisons of the

model to observations are presented after averaging the observations along the flight tracks over the model grid.

Specific KORUS-AQ aircraft measurements used in this work include thermal dissociation laser-induced fluorescence (TD-LIF) for $NO_2$ concentrations (Thornton 2000; Nault et al., 2015), NCAR 4-Channel $NO_{xy}O_3$ Chemiluminescence for NO and $O_3$ concentrations (Walega et al. 1991), NCAR CCD-based Actinic

Flux Spectrometer (CAFS) for $NO_2$ photolysis frequencies ($j_{NO_2}$), airborne tropospheric hydrogen oxides sensor (ATHOS) for OH and $HO_2$ concentrations and OH reactivity (OHR) (Faloona et al., 2004; Mao et al., 2009), Differential Absorption Carbon monOxide Measurement (DACOM) for CO concentrations (Sachse et al., 1987), A high-Resolution Time-of-Flight Aerosol Mass Spectrometer (CU HR-ToF-AMS) for sulfate, nitrate, and ammonium aerosols (DeCarlo et al., 2006; Nault et al., 2018) and a $CF_3O^-$ Chemical Ionization

Time of Flight Mass Spectrometer (CIT-ToF-CIMS) for $H_2O_2$ concentrations (Crounse et al., 2006). Both TD-LIF and $NO_{xy}O_3$ Chemiluminescence measured $NO_2$ concentrations, and both are known for positive interferences at high altitudes from $CH_3O_2NO_2$ and $HNO_4$ dissociation (Reed et al., 2016; Nussbaumer et al., 2021). Since TD-LIF data attempts to correct positive interference and detect $NO_2$ directly (Nault et al., 2015), TD-LIF $NO_2$ is used for this work. We also use surface $NO_2$ concentrations from a Teledyne T500U CAPS

analyzer located at Olympic Park in the SMA.

### 2.2. GEOS-Chem model



We use GEOS-Chem version 13.3.4 (https://doi.org/10.5281/zenodo.5764874, last accessed 20 March 2022)
driven by the Goddard Earth Observing System - Forward Processing (GEOS-FP) assimilated meteorological
data with a horizontal resolution of 0.25° × 0.3125° and 72 vertical levels up to the mesopause including 14
levels below 2 km altitude. The native resolution is used over the Korean peninsula and surrounding ocean
(29° - 40°N, 120° - 135°E) with dynamic boundary conditions updated every three hours from a global 4° ×
5° simulation. The simulation is conducted for April-June 2016 following 6 months of initialization.
Emissions in GEOS-Chem are calculated by the Harmonized Emissions Component (HEMCO) (H. Lin et
al., 2021). Global anthropogenic emissions in GEOS-Chem are from the Community Emissions Data System
(CEDSv2) (McDuffie et al., 2021), superseded in China by the Multi-resolution Emission Inventory (MEIC;
Zheng et al., 2018), and for the rest of Asia including shipping by the KORUSv5 inventory (Woo et al., 2020).
We apply a diurnal profile of $NO_x$ emission rates derived from an emission inventory for SMA (Goldberg et
al., 2019). Natural emissions are from Guenther et al. (2012) for biogenic VOCs (MEGANv2), Murray et al.
(2012) for lightning $NO_x$, Hudman et al. (2012) for soil $NO_x$, and Jaeglé et al. (2011) for sea-salt aerosol
(SSA) as $Na^+$. GEOS-Chem version 13.3.4 (v13.3.4) includes a detailed oxidant-aerosol chemical mechanism
(Wang et al., 2019; R. Park et al., 2004) with recent updates for halogen chemistry (Wang et al., 2021), $NO_x$
heterogeneous and cloud chemistry (Holmes et al., 2019), isoprene chemistry (Bates and Jacob, 2019), and
aromatic chemistry (Bates et al., 2021). We introduce additional updates as described below.

Several previous studies have used GEOS-Chem to simulate and interpret KORUS-AQ observations. Choi
et al. (2019) studied local and transboundary sources of fine particulate matter ($PM_{2.5}$). Zhai et al. (2021)
analyzed aerosol vertical profiles for the interpretation of satellite aerosol optical depth (AOD) observations.
Travis et al. (2022) found a large model overestimate of nitrate and underestimate of sulfate aerosols. R. Park
et al. (2021) showed systematic low biases in simulated CO and $O_3$ concentrations at all altitudes. Oak et al.
(2019) found that correcting the daytime planetary boundary layer (PBL) height and increasing $NO_x$ emission
(since corrected in the KORUSv5 inventory) led to a more accurate simulation of surface $O_3$ concentration.
H. Kim et al. (2022) found that simulated OHR was too low and attributed this to underestimates of CO and
oxygenated VOC (OVOC) emissions. Model comparison to formaldehyde (HCHO) observations from the
aircraft also found an underestimate in emissions of anthropogenic VOCs (aVOCs) (Kwon et al., 2021).

Here we make four updates to the standard GEOS-Chem model to better represent the oxidant chemistry over
South Korea: (1) addition of $NO_3^-$ photolysis, (2) uptake of $HNO_3$ by coarse anthropogenic dust, (3) VCP
emissions, and (4) $HO_2$ heterogeneous uptake by aerosols. We describe each of them in order.

We incorporate $NO_3^-$ photolysis in the model following Shah et al. (2022). Observations in marine air show
high nitrous acid (HONO) concentrations that have been attributed to $NO_3^-$ photolysis with support from



laboratory experiments in particular for Cl-containing aerosol (Zhou et al., 2008; Ye et al., 2016; Reed et al.,
2017). Kasibhatla et al. (2018) initially added $NO_3^-$ photolysis in coarse sea salt aerosol (SSA) as an option
in GEOS-Chem. Shah et al. (2022) extended this to all $NO_3^-$ aerosols to correct a low model bias for $NO_x$
concentrations in the remote troposphere. The $NO_3^-$ photolysis frequency ($j_{NO_3^-}$) is computed as an
enhancement factor (EF) relative to the $HNO_3$ photolysis frequency ($j_{HNO_3}$): $j_{NO_3^-} = EF \times j_{HNO_3}$. An EF of
100 is applied for coarse-mode (>1 μm) sea-salt $NO_3^-$ aerosol. For fine mode $NO_3^-$ aerosol, which in GEOS-
Chem is an internal mixture of sulfate-nitrate-ammonium (SNA) and fine sea-salt, the EF is calculated using
equation (1):

$$EF = \max\left(100 \times \frac{1}{1 + \frac{[NO_3^-]}{[SSA]}}, \ 10\right) \tag{1}$$

where [ ] denotes a molar concentration and $[SSA] = 2.39[Na^+]$ ($Na^+$ is the inert SSA tracer in GEOS-Chem).
Our resulting mean EF value for the fine mode $NO_3^-$ aerosol along the KORUS-AQ flight tracks is 15. Romer
et al. (2018) inferred a non-SSA EF value in the range of 1-30 from the ratio of $NO_x$ to total inorganic nitrate
measured in KORUS-AQ over the Yellow Sea.

We add $HNO_3$ uptake by anthropogenic coarse particulate matter ($PM_{10} - PM_{2.5}$) following Zhai et al. (2022a).
Coarse particulate matter measured by the surface air quality network in South Korea is very high, averaging
20 μg m⁻³, and most are anthropogenic fugitive dust that would take up $HNO_3$ similarly to natural dust (Fairlie
et al., 2010). Zhai et al. (2022a) found that they could correct in this manner the previously identified high
model biases for $HNO_3$ and $NO_3^-$ during KORUS-AQ in the daytime though there remains nighttime
overestimate (Zhai et al., 2021; Travis et al., 2022).

We implement VOC emissions from non-industrial uses of volatile chemical products (VCPs) (e.g.,
adhesives, cleaning agents, coatings, pesticides, printing inks, and personal care products) based on per capita
emission estimates from McDonald et al. (2018) and Coggon et al. (2021), as incorporated into GEOS-Chem
by Bates et al. (2022). The KORUSv5 inventory does not account for non-industrial VCP emissions but
observations in the SMA suggest that they are important (S. Kim et al., 2016; Simpson et al., 2020), and we
find in GEOS-Chem that they account for 97% of ethanol anthropogenic emissions, 94% of acetone, 31% of
toluene, and 90% of methanol. The VCPs are emitted following a diurnal emission profile from Coggon et
al. (2021). Oxidation of VCP ethanol is a major source of acetaldehyde for the conversion of $NO_x$ to
peroxyacetylnitrate (PAN) (Zhai et al., 2022b). Oak et al. (2019) and Schroeder et al. (2020) found toluene
to dominate $O_3$ production in the SMA. Toluene emission increases by 44% when including VCPs.



Finally, we decrease the reactive uptake coefficient $\gamma$ of $HO_2$ ($\gamma_{HO_2}$) by aerosols to 0.1, as compared to 0.2 in the standard GEOS-Chem model (Martin et al., 2003; Mao et al., 2010). As we will show in Section 3, $\gamma_{HO_2}$=

0.1 gives a better fit to the $HO_2$ observations and their relationship to aerosol concentrations in KORUS-AQ. Brune et al. (2022) used a box model to show that $\gamma_{HO_2}$= 0.2 is too high for simulating $HO_2$ observations in KORUS-AQ. A $\gamma_{HO_2}$ value of 0.1 is within the range of current knowledge (Lakey et al., 2015; Zou et al., 2019; Taketani et al., 2012). Reactive uptake of $HO_2$ by aerosol can have either $H_2O$ or $H_2O_2$ as a product, and GEOS-Chem assumes $H_2O$ by default to avoid overestimation of $H_2O_2$ observations (Mao et al., 2010,

2013). This is supported for the KORUS-AQ conditions by previous model studies finding that the assumption of $H_2O_2$ as a product of $HO_2$ heterogeneous uptake leads to a large overestimate of observed $H_2O_2$ concentrations (Miyazaki et al., 2019; Gaubert et al., 2020).

Underestimation of CO in the northern hemisphere is a common problem for CTMs (Shindell et al., 2006; Huijnen et al., 2010) and this was also found for GEOS-Chem in KORUS-AQ (R. Park et al., 2021; H. Kim

et al., 2022). Here we simply implement a 50% increase in the GEOS-Chem CO concentrations used as boundary conditions to overcome this underestimation.

## 2.3. Air Mass Factor (AMF) Calculation

Satellite retrieval of the tropospheric $NO_2$ VCD $\Omega_v$ (molecules $cm^{-2}$) applies an AMF for the scattering atmosphere to the tropospheric $NO_2$ SCD $\Omega_s$ measured along the line of sight of the instrument by the spectral

fitting of the backscattered radiances and subtracting the stratospheric contribution:

$\Omega_v = \Omega_s / AMF$                                               (2)

The AMF depends on the viewing geometry, the scattering properties of the surface and the atmosphere, and the vertical distribution of $NO_2$, as given by Palmer et al. (2001):

$AMF = AMF_G \int_0^{z_T} w(z)S(z)\mathrm{d}z$                                               (3)

Here $AMF_G$ is the geometric AMF assuming no atmospheric scattering and is a function of solar zenith angle (SZA) and viewing zenith angle (VZA); $w(z)$ is a scattering weight that measures the relative sensitivity of the backscattered radiance to the $NO_2$ optical depth at altitude $z$, averaged over the wavelength range of the spectral fitting window (Lamsal et al., 2021); $S(z) = n(z)/\Omega_v$ is a shape factor for the vertical distribution of the $NO_2$ number density $n(z)$; the integration is from the surface to the tropopause $z_T$. The scattering weights

are such that $w(z) = 1$ for all $z$ in a non-scattering atmosphere, in which case $AMF = AMF_G$ is solely defined by the geometric light path from the Sun to the satellite as reflected by the surface. We will refer to $\int_0^{z_T} w(z)S(z)\mathrm{d}z$ in what follows as the scattering correction factor.





Here we use shape factors from the KORUS-AQ vertical profiles, either observed from the aircraft or simulated with GEOS-Chem, to compute the AMF. The scattering weights are obtained from the OMI $NO_2$

retrieval lookup table (LUT) as a function of SZA, VZA, relative azimuth angle (RAA), surface pressure, and albedo (Bucsela et al., 2006; Bucsela et al., 2013). We compute SZA, VZA, and RAA for the locations and times of the KORUS-AQ aircraft profiles and the geostationary location of the GEMS satellite instrument (128°E longitude, 0° latitude, 35786 km mean altitude). We assume a surface pressure of 1013 hPa. The surface albedo is determined from OMI Level-3 LER climatology (OMLER; Kleipool et al., 2008) for June

and the locations of the KORUS-AQ profiles.

Our specific interest in computing AMFs for the KORUS-AQ conditions is for application to GEMS retrievals. The spectral retrieval windows for OMI (402 – 465 nm) and GEMS (432 – 450 nm) have a sufficient overlap that we can assume the scattering weights to be similar for our analysis, considering that the wavelength dependence of scattering weights is weak (Palmer et al., 2001). We will limit our attention to

clear-sky scenes. Retrieval for partly cloudy scenes in the GEMS algorithm uses a cloud albedo (0.8) instead of the surface albedo for the cloudy fraction of the scene, assumes $w(z) = 0$ below cloud top, and weighs the clear-sky and cloudy scattering weights by the relative contributions to the back-scattered radiances (Martin et al., 2002; Lee et al., 2020). Our focus on clear-sky scenes does not detract from the generality of our results, since the extension to partly cloudy scenes is straightforward, except for the possible role of cloud in altering

the $NO_2$ vertical profile but this is generally not considered in retrieval algorithms.

## 3. Results and Discussion

### 3.1. Oxidant chemistry and $NO_x$ vertical profiles during KORUS-AQ

Figure 1 shows the relationship between $HO_2$ and sulfate-nitrate-ammonium (SNA) aerosol concentrations as observed in KORUS-AQ and simulated by GEOS-Chem. The model with $\gamma_{HO_2} = 0.2$ shows depression of

$HO_2$ concentrations at high aerosol concentrations when uptake by aerosol is the dominant $HO_x$ sink but that depression is not seen in the observations. The model with $\gamma_{HO_2} = 0.1$ corrects this behavior. Figure 2 shows the median simulated (red line) and observed (black line) vertical profiles of oxidants and related quantities in the SMA during KORUS-AQ, highlighting the effects of model updates over the standard GEOS-Chem v13.3.4 (blue line). Results outside the SMA are in Figure S1 and show similar behavior.


We see from Figure 2 that the low $O_3$ background bias previously reported by R. Park et al. (2021) is largely corrected, mainly from the inclusion of $NO_3^-$ photolysis in the global simulation (Shah et al., 2022). The simulated $O_3$ enhancement in the PBL below 2 km matches observations much better than GEOS-Chem





v13.3.4 and this is mostly driven by higher $HO_2$ concentrations, with some additional contribution from VCPs
and higher CO boosting the OHR. The higher $HO_2$ concentrations relative to GEOS-Chem v13.3.4 are due
to a combination of slower heterogeneous uptake, the addition of VCP emissions, feedback from increasing
$O_3$, and an increase in CO concentrations. Sensitivity simulations show that each of these factors contributes
a $1 - 2$ pptv increase in $HO_2$ concentration. An increase in $HO_2$ concentration drives an increase in $H_2O_2$
concentration, which was previously underestimated by 21% in the PBL and is now overestimated by 42%.
An increase in the $H_2O_2$ deposition velocity could reconcile the model and observations (Allen et al., 2022).
Simulated OH concentrations are 31% higher than observed, which is still within the accuracy of the
measurement; there is little difference between our simulation and GEOS-Chem v13.3.4 for OH because
higher $HO_x$ concentrations are offset by an increase in the OHR. The OHR is still lower than observed (H.
Kim et al., 2022).

Figure 3 shows the median simulated (red line for GEOS-Chem; blue line for GEOS-Chem v13.3.4) and
observed (black line) vertical profiles of NO and $NO_2$ concentrations in the SMA, together with the $NO/NO_2$
concentration ratios. The same plot outside the SMA is shown in Figure S2 and the conclusions regarding
the $NO/NO_2$ ratio are similar. NO and $NO_2$ are mainly in the PBL. The model underestimates concentrations
slightly in the SMA, which could have to do with the 25-km spatial resolution since that underestimate is not
seen outside the SMA. The $NO/NO_2$ ratio is expected to be governed in the daytime by a photostationary
steady state (PSS) between the fast oxidation of NO and photolysis of $NO_2$:

$$NO + O_3 \rightarrow NO_2 + O_2 \qquad (R1)$$
$$NO + HO_2 \rightarrow NO_2 + OH \qquad (R2)$$
$$NO + RO_2 \rightarrow NO_2 + RO \qquad (R3)$$
$$NO + XO \rightarrow NO_2 + X \qquad (R4)$$
$$NO_2 + h\nu \xrightarrow{+O_2} NO + O_3 \qquad (R5)$$

where $RO_2$ denotes organic peroxy radicals and X denotes halogens. PSS is then defined by

$$PSS = \frac{[NO]}{[NO_2]} = \frac{j_{NO_2}}{k_{NO+O_3}[O_3] + k_{NO+HO_2}[HO_2] + \sum k_{NO+RO_2}[RO_2] + \sum k_{NO+XO}[XO]} \qquad (4)$$

where [ ] denotes number density, $k$ denotes rate constants (Burkholder et al., 2020), and the summation $\Sigma$ is
applied to all $RO_2$ or XO species. Figure 3 shows the PSS (gold line) computed from aircraft measurements
except for [XO] and [$RO_2$] which are from GEOS-Chem but contribute generally less than 4% of the
conversion from NO to $NO_2$.

We see from Figure 3 that the $NO/NO_2$ ratio increases with altitude, which largely reflects the temperature
dependence of the $NO + O_3$ reaction (Burkholder et al., 2020). The model closely tracks the PSS, but the





observations deviate above 5 km. This may be attributed to positive artifacts in the TD-LIF $NO_2$ instrument
at low temperatures (Shah et al., 2022). This implies that the PSS assumption applied to the more accurate
NO concentrations provides a better estimate of $NO_2$ concentrations above that altitude. Hereafter, we replace
the $NO_2$ observations above 5 km altitude with values inferred from the observed NO concentrations and
PSS.

**3.2. Air mass factors for satellite $NO_2$ retrievals and their diurnal variation**

Figure 4 shows the median vertical profile of simulated (red line) and observed (black line) $NO_2$ number
density in the SMA during KORUS-AQ, along with the scattering weights for satellite retrievals and the
cumulative vertical contributions to the tropospheric slant column as would be detected from a satellite
instrument. No observations are available above 7 km and GEOS-Chem is used there instead. Observed $NO_2$
between 5 and 7 km is inferred from NO observations using PSS. The clear-sky scattering weights (left panel)
represent the sensitivity of the instrument to $NO_2$ as a function of altitude and increase by a factor of 2.4 from
the surface to 6 km altitude. The fractional cumulative contribution to the tropospheric slant column density
($\Omega_s$ in equation (2)) is calculated as $\int_0^z w(z)n(z)\mathrm{d}z$ / $\int_0^{z^T} w(z)n(z)\mathrm{d}z$ and represents the fraction of the
measured column contributed by $NO_2$ below altitude $z$. $NO_2$ in the PBL below 2 km contributes 95% of the
SCD over the SMA and 80% of the SCD for the ensemble of KORUS-AQ observations, reflecting the highly
polluted conditions. Previous work over the southeastern US in the summer found that the PBL contributed
only 20-35% of the SCD (Travis et al., 2016), such that the large contribution from the free tropospheric
background had to be considered when inferring $NO_x$ emissions from the $NO_2$ columns (Silvern et al., 2019;
Qu et al., 2021). This is much less of an issue for South Korea. The AMF inferred from the median observed
$NO_2$ vertical profile in the SMA during KORUS-AQ is 1.18, in close agreement with the corresponding value
of 1.22 from GEOS-Chem.

GEMS offers the first opportunity to directly observe the diurnal variation of $NO_2$ from space, but it is
important to understand how much the AMF contributes to this diurnal variation and how well this is resolved
by the model used to obtain shape factors. During the KORUS-AQ campaign, repeated daily flights in the
morning (8 – 9 LT), midday (12 – 13 LT), and afternoon (15 – 16 LT) were conducted on 10 flight days over
two supersites: Olympic Park (37.5232 °N, 127.1260 °E) and Mt. Taewha (37.3123 °N, 127.3106 °E)
(Crawford et al., 2021). Olympic Park is in the SMA, and Mt. Taewha is a research forest site 29 km southeast
of Olympic Park. Mt. Taewha is affected by SMA air pollution through downwind transport. The flight
patterns involved missed landing approaches to very low flight altitudes over Olympic Park followed by
spiral ascents to the east of Taewha up to 7 km altitude (Crawford et al., 2021). To construct the diurnal
variation of the $NO_2$ vertical profile over the SMA and its implication for the AMF, we combine the Olympic





Park vertical profiles (typically $z < 2$km) with the spirals east of Mt. Taewha. A similar approach was taken to investigate HCHO AMF using KORUS-AQ campaign data (Spinei et al., 2018). We also use the surface $NO_2$ data from Olympic Park.

Figure 5 shows the clear-sky scattering weights at 8 – 9 LT (orange line), 12 – 13 LT (purple line), and 15 – 16 LT (light blue line). In addition to the dependence on altitude, the scattering weights show a dependence on the time of day driven by the solar zenith angle (SZA) and relative azimuth angle (RAA), such that the sensitivity to near-surface $NO_2$ at 8 – 9 LT is 14% lower than at 12 – 13 LT. Figure 5 also shows the clear-sky median vertical distributions of simulated (red line) and observed (black line) $NO_2$ number densities at 8

– 9 LT, 12 – 13 KT, and 15 – 16 LT. The $NO_2$ vertical profiles follow the rise in the mixed layer from early morning to early afternoon in response to surface heating, with higher near-surface concentrations in the morning when the mixed layer is shallow and lower surface concentrations in the afternoon due to deeper vertical mixing. Observations during KORUS-AQ indicated a mean mixed layer height at 8 – 9 LT of about 0.6 km rising to about 1.7 km in the early afternoon (Travis et al., 2022). Model concentrations below 1 km

altitude are lower than observed, which could reflect the local urban nature of the site as compared to the 25-km grid resolution of the model.

Although surface $NO_2$ peaks strongly in the morning (Figure 5), the $NO_2$ VCDs observed from the ground-based sun-staring column measurements show an increase from early morning to 11 LT, steady concentrations until 14 LT, and a slight decrease afterward (Chong et al., 2018; Crawford et al., 2021).

Vertical mixing would not directly affect the column and the increase of the column over the morning could reflect the effect of accumulating emissions. An important difference between the ground-based and satellite measurements of $NO_2$ columns is that there is no vertical dependence of $NO_2$ sensitivity for the former.

Table 1 shows the AMF diurnal variation inferred from the $NO_2$ vertical profiles in Figure 5 for the three different times of the day. This diurnal variation is driven by three factors: 1) the dependence of the light path

on SZA ($AMF_G$), 2) the dependence of the scattering weights $w(z)$ on SZA and RAA, and 3) the dependence of the shape factors $S(z)$ on the diurnal cycle of mixed layer growth. The scattering correction factor $\int_0^{z_T} w(z)S(z)\mathrm{d}z$ in Table 1 captures the combined effects of $w(z)$ and $S(z)$. From a purely geometric perspective (non-scattering atmosphere), as measured by $AMF_G$, the measurement would be 28% more sensitive in the morning than at midday because of the longer light path. However, this effect is offset by the scattering

correction factor which is 17% smaller in the morning than at midday, because $NO_2$ in the morning is closer to the surface and therefore harder to detect. As a result, the AMF is only 7% higher in the morning than at midday. There is no such offsetting effect from midday to afternoon since the $NO_2$ vertical profiles are similar in midday and afternoon. Thus, the AMF in the afternoon is 14% larger than at midday.



The relative diurnal variations of the tropospheric $NO_2$ VCDs (22%; Crawford et al., 2021) and of the profile-
driven scattering correction factor (21%) are of comparable relative magnitude, indicating that the diurnal
variation in the $NO_2$ vertical profile affecting the AMF is important to account for when interpreting the
diurnal variation of VCDs from the satellite. We see from Table 1 that GEOS-Chem reproduces closely the
observed diurnal variation of the scattering correction factor and hence the AMF.

We investigate more broadly in Figure 6 the observed variability of the AMF, and the ability of GEOS-Chem
to reproduce it, for the 63 vertical profiles collected during KORUS-AQ. These include 41 vertical profiles
that combine flights over Olympic Park and Mt. Taewha spirals at different times of day, 17 vertical profiles
that combine flights over Osan (just south of Seoul) and Mt. Taewha spirals, and 5 vertical profiles over the
ocean. All individual profiles extend from below 0.5 km to above 6.5 km. For the Osan and ocean profiles,
we extend the aircraft observations from the lowest altitude to the surface as there are no surface
measurements. Oceanic data is shown in navy, and the land data is colored depending on the time of the day
(7 – 9 LT in orange, 11 – 13 LT in purple, and 14 – 17 LT in light blue).

The observed AMF in Figure 6 ranges from 1.05 to 1.63, representing a major factor of variability in the
retrieved tropospheric $NO_2$ VCDs. The two main drivers of this observed AMF variability are surface type
(ocean versus land), and time of day for land. Oceanic profiles have higher AMF (1.55±0.068) than land
(1.23±0.097) because the $NO_2$ PBL enhancement is weaker or absent (Martin et al., 2002). All KORUS-AQ
vertical profiles over land have large $NO_2$ PBL enhancements, leading to the time of day being the driving
factor of variability as previously discussed. GEOS-Chem captures 53% of the variance in the observed AMF
for individual profiles, and its overall error in computing the AMF is relatively small with a normalized mean
bias (NMB) of 2.7% and a relative root-mean-square error (RRMSE) of 7.6%. The largest discrepancies are
for four early-morning profiles (7 – 9 LT), which are attributable to a model error in the timing of mixed
layer growth (Travis et al., 2022), being either too early (AMF overestimate) or too late (AMF underestimate).

The AMF errors from using GEOS-Chem model $NO_2$ shape factors, as shown in Figure 6, can be placed in
the context of other contributions to the AMF error. Boersma et al. (2007) estimated that the AMF contributes
a 30% error to OMI tropospheric $NO_2$ VCD retrievals, with dominant contributions from uncertainties in
clouds and surface albedo. Uncertainty in the shape factors as computed from the TM4 CTM accounted for
30% of the AMF error or 9% of the overall error in the tropospheric $NO_2$ VCD. Our RRMSE of 7.6% from
the use of GEOS-Chem shape factors to compute the AMF under KORUS-AQ conditions is consistent with
their results and extends them to observations at different times of the day. Considering that the satellite
observations are typically averaged over many days for analysis, which would reduce the RRMSE, a more
important consideration is the NMB which we find to be only 2.7% with GEOS-Chem. Of most concern is





that the interpretation of the diurnal variability of tropospheric $NO_2$ VCDs from space could be affected by large AMF errors in the morning hours due to incorrect model timing of mixed layer growth. In GEOS-Chem at least, this error does not appear to be systematic so averaging over multiple days would dampen it.

Our analysis focused on clear-sky conditions. However, it would be rare to find a 7×8 km$^2$ GEMS pixel
without clouds (Remer et al., 2012). Clouds may dominate the AMF error budget for cloud fractions larger than 20% corresponding to more than 50% of the observed radiance originating from the cloudy fraction (Boersma et al., 2007). The GEMS algorithm (Lee et al., 2020) computes the AMF for cloudy conditions by assuming scattering weights of zero below the cloud top. The AMF for a partly cloudy scene with radiance-weighted cloud fraction $f$ is then obtained as

$$AMF = AMF_G \left( f \int_{z_c}^{z_T} w_c(z)S(z)\mathrm{d}z + (1-f) \int_0^{z_T} w(z)S(z)\mathrm{d}z \right) \tag{5}$$

where $z_c$ is the cloud top altitude and the scattering weights $w_c$ above the cloud top are computed using the cloud albedo. Although clouds would be expected to affect the $NO_2$ vertical profile through photolysis, chemistry, and vertical motions, these effects are complex, and they are not an obvious source of error when comparing model and observed $NO_2$ profiles. From a diurnal variability perspective, the afternoon formation
of fair-weather cumuli would decrease the sensitivity of the satellite measurement to the PBL and therefore alias the observed diurnal variation of $NO_2$. The exploitation of GEMS data to interpret diurnal variations of tropospheric $NO_2$ VCDs in terms of emissions or chemistry will need to focus on conditions with small cloud fractions.

## 4. Conclusions

We used extensive observations of $NO_2$ vertical profiles and related chemistry from the KORUS-AQ aircraft campaign over and around South Korea in May-June 2016 to better understand how the vertical distribution of $NO_2$ affects the air mass factors (AMFs) for satellite retrievals of tropospheric $NO_2$ vertical column density (VCD) in the polluted East Asia chemical environment. This was motivated by the recent launch of the GEMS geostationary instrument over East Asia, enabling the first-time direct measurements of the diurnal
variation of $NO_2$ from space, and by interest in using the GEOS-Chem chemical transport model (CTM) to provide the $NO_2$ relative vertical profiles (shape factors) needed for the retrievals. We examined more broadly the role of oxidant chemistry in controlling $NO_2$ concentrations in GEOS-Chem through the photostationary steady state (PSS) $NO/NO_2$ ratio and the lifetime of $NO_x$.

We introduced several updates to GEOS-Chem to improve the model representation of oxidant chemistry
over East Asia, drawing on previous evaluations of the standard version of the model with the KORUS-AQ data. The inclusion of aerosol nitrate photolysis in the model corrects the previous underestimate of the $O_3$



background. The standard model depleted $HO_2$ under highly polluted conditions due to uptake by aerosol particles but this was not seen in observations. Decreasing the $HO_2$ reactive uptake coefficient corrects this model behavior, and model $HO_2$ is further brought into agreement with observations by the addition of

volatile chemical products (VCPs) chemistry and an increase in CO. The $HO_2$ correction in turn allows the model to simulate the observed $O_3$ enhancement in the planetary boundary layer (PBL) below 2 km altitude. Increasing $O_3$ and $HO_2$ in the model improves the simulation of the PSS $NO/NO_2$ ratio.

The KORUS-AQ vertical profiles indicate that 95% of the tropospheric $NO_2$ VCD detected from space over the Seoul Metropolitan Area (SMA) originates from the PBL, despite the much higher sensitivity to $NO_2$ in

the free troposphere, reflecting the highly polluted conditions. This leads to a large diurnal variability of $NO_2$ detected from space that can be interpreted by GEMS in terms of $NO_x$ emissions and chemistry but requires correct accounting of the diurnal variation in AMF. KORUS-AQ offered a unique dataset for addressing this issue with repeated diurnal vertical profiles over SMA at different times of the day. We find that the diurnal evolution of the $NO_2$ shape factor resulting from mixed layer growth drives a 21% increase in the scattering

correction factor from 8 – 9 LT to 12 – 13 LT, offsetting the geometric decrease in the resulting AMF from the shorter light path. GEOS-Chem can capture this diurnal variability in the shape factor driven by mixed layer growth with no significant mean error. Incorrect timing of mixed layer growth can lead to large errors for individual morning profiles but averaging over a number of days dampens the error.

The AMF values inferred from the collection of KORUS-AQ vertical profiles range from 1.05 to 1.63, with

most of that variability driven by ocean versus land and by the time of day over land. GEOS-Chem captures 53% of this variability in the AMF with a relative root-mean-square error (RRMSE) of 7.6% and normalized mean bias (NMB) of 2.7%. These errors are relatively small compared to other sources of retrieval errors and support the use of GEOS-Chem vertical profiles in GEMS retrievals.

### *Code Availability*

The model code used in this work is available at https://doi.org/10.5281/zenodo.5764874.

### *Data Availability*

The KORUS-AQ data archive is available at https://www-air.larc.nasa.gov/ and it includes both the ground-based and aircraft-based measurements (KORUS-AQ Science Team, 2019).

### *Author Contribution*

The original draft preparation was done by LHY, with review and editing by DJJ, KRT, JHC, JHK, and JK. DJJ contributed to project conceptualization. Modeling was done by LHY, with additional support from NKC,



SZ, KHB, VS, EB, RMY, HL. The formal analysis was conducted by LHY with additional support from DJJ, JFB, HC, KRT, JC. The LUT used for the scattering weights is provided by LL.

*Competing Interests*

The contact author has declared that none of the authors has any competing interests.

*Acknowledgements*

This work was funded by the Samsung Advanced Institute of Technology. The authors would like to acknowledge KORUS-AQ data providers including Ronald Cohen for TD-LIF, Andrew Weinheimer for 450 $NO_{xy}O_3$ Chemiluminescence, Samuel Hall for CAFS, William Brune for ATHOS, Glenn Diskin for DACOM, Jose Jimenez for AMS, Paul Wennberg for CIT-CIMS, and Russell Long for CAPS measurements.

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





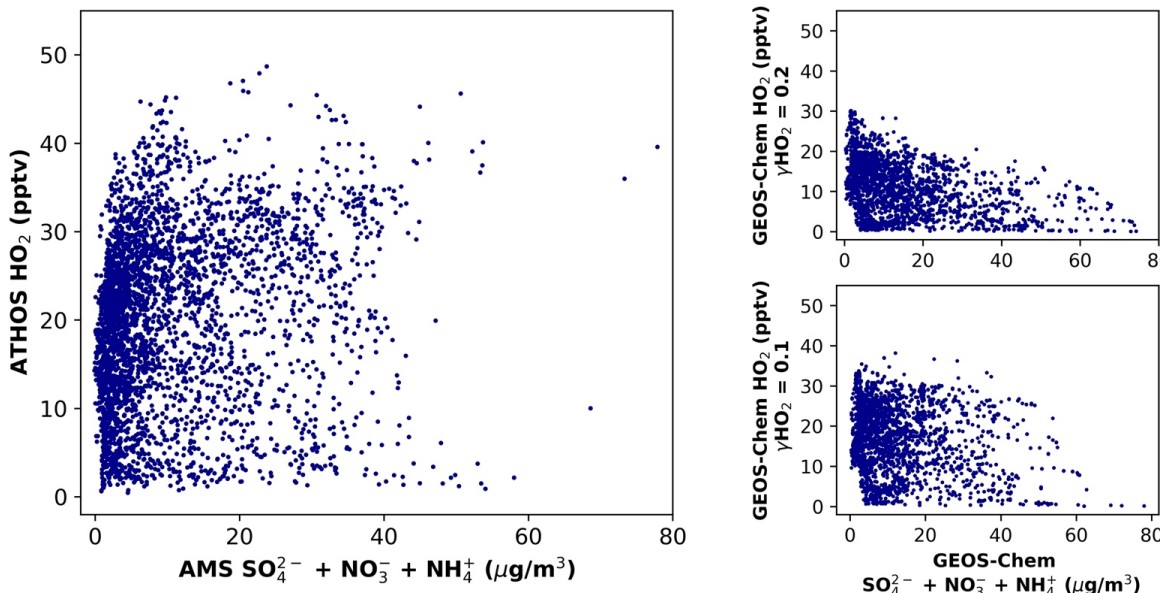

**Figure 1.** Relationship of HO$_2$ and sulfate-nitrate-ammonium (SNA) concentrations below 2km altitude
during KORUS-AQ. The left panel shows the ensemble of observations from the ATHOS and AMS
instruments. The right panels show GEOS-Chem model results with HO$_2$ reactive uptake coefficients($\gamma_{HO_2}$)
of 0.2 (top panel) and 0.1 (bottom panel).







**Figure 2**. Median vertical profiles of species concentrations and OH reactivity (OHR) in the Seoul Metropolitan Area (SMA; 37 – 37.6°N, 126.6 – 127.7°E) during KORUS-AQ. Observations are compared to our GEOS-Chem simulation and the standard version 13.3.4 of the model.



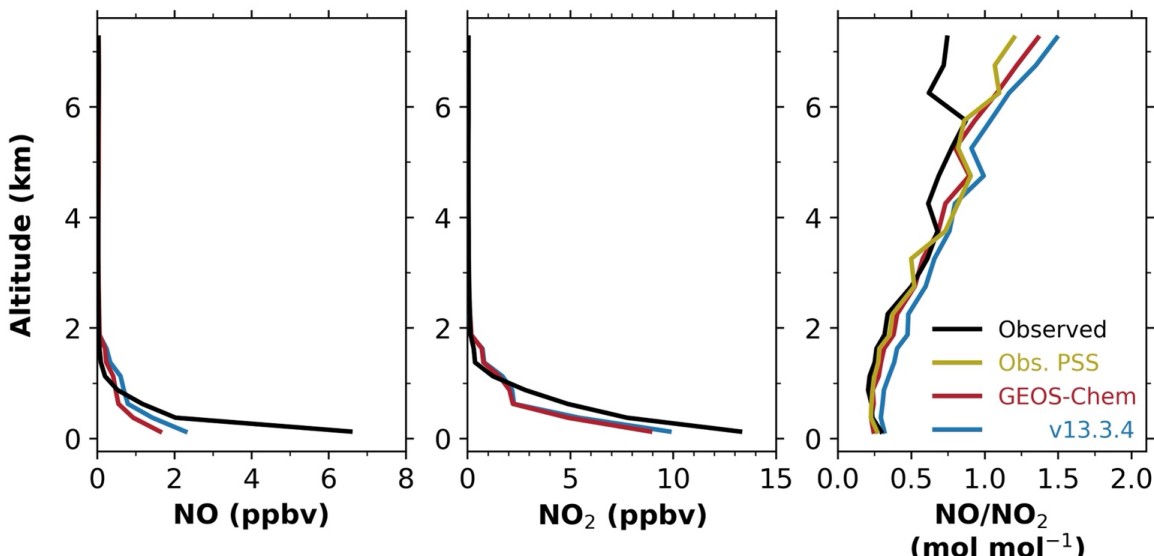

**Figure 3**. Median vertical profiles of NO and $NO_2$ concentrations, and $NO/NO_2$ concentration ratios, in the
SMA (37 – 37.6°N, 126.6 – 127.7°E) during the KORUS-AQ campaign. Observations are compared to our
GEOS-Chem simulation and the standard version 13.3.4 of the model. PSS for the $NO/NO_2$ ratio denotes a
photochemical steady state as given by equation (4) and is computed mainly from observed quantities.
Observed PSS is computed only if both species are more than 2× above the limit of detection.



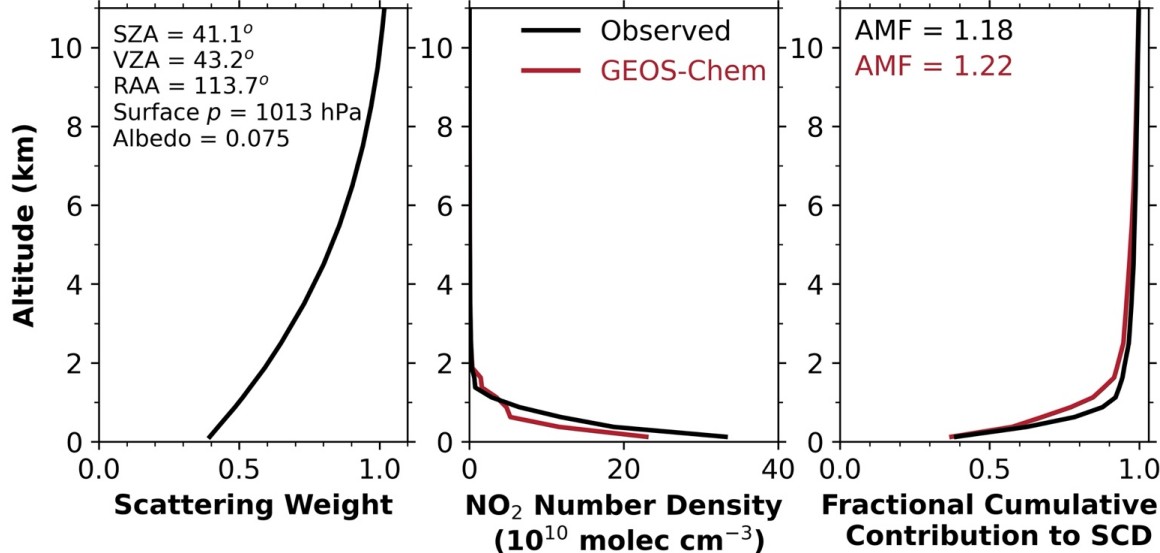


**Figure 4**. Vertical profile contributions to the tropospheric NO$_2$ columns that would be detected from a satellite instrument. The left panel shows the mean clear-sky scattering weights ($w(z)$ in equation (3)) for the ensemble of observed vertical profiles over the SMA during KORUS-AQ. The scattering weights represent the altitude-dependent sensitivity of the detected slant column to NO$_2$ optical depth, which scales with

number density. The middle panel shows observed and simulated median vertical profiles of NO$_2$ number density in the SMA during KORUS-AQ. The right panel shows the fractional cumulative contribution to the NO$_2$ slant tropospheric column density (SCD; $\Omega_s$ in equation (2)) from NO$_2$ below a given altitude in the SMA. The air mass factors (AMF; equation (3)) given inset are obtained by applying the scattering weights from the left panel to the shape factors from the middle panel and including a mean geometric AMF$_G$ of 2.70.

Observed NO$_2$ number densities between 5 and 7 km are inferred from NO observations and applying PSS (equation (4)). No observations were made above 7 km and GEOS-Chem values are used there instead.



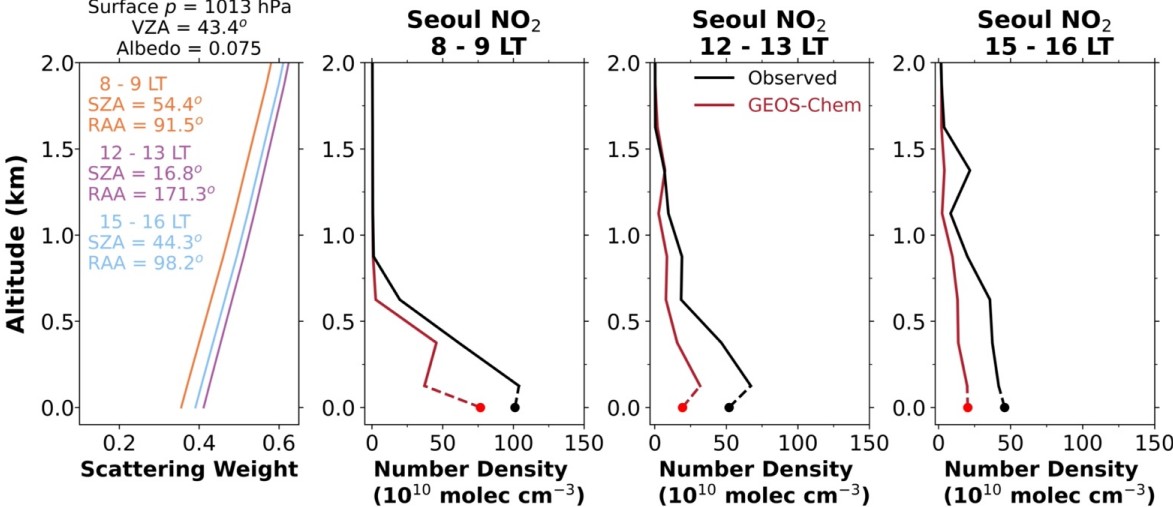

**Figure 5.** Clear-sky scattering weights for tropospheric NO$_2$ column retrievals over Olympic Park in Seoul
at different times of day with the GEMS viewing geometry (left) and median vertical profiles of NO$_2$ number
densities at different times of the day. Observations are from 10 KORUS-AQ flights that repeated the same
flight pattern of vertical profiling under clear-sky conditions (May 4, 7, 17, 18, 30, 31; June 2, 3, 9, 10) at 8
– 9 local time (LT), 12 – 13 LT, and 15 – 16 LT. SZA and RAA are averaged over the corresponding vertical
profiles. The clear-sky scattering weights are taken from the OMI TOMRAD look-up table. The surface NO$_2$
concentration shown as a symbol is from the Teledyne T500U CAPS analyzer located at Olympic Park station.

Table 1. Diurnal variation of the air mass factor (AMF) for satellite NO$_2$ retrievals[1]

| Time of day[2] | SZA | RAA | AMF$_G$ | $\int_{0}^{z_T} w(z)S(z)\mathrm{d}z$ | AMF |
|---|---|---|---|---|---|
| 8-9 LT | 54.4° | 91.5° | 3.09 | 0.38 (0.39) | 1.19 (1.20) |
| 12-13 LT | 16.8° | 171.3° | 2.42 | 0.46 (0.47) | 1.11 (1.14) |
| 15-16 LT | 44.3° | 98.2° | 2.77 | 0.46 (0.46) | 1.27 (1.28) |

[1] Inferred from the median observed vertical profiles of NO$_2$ concentrations over Seoul taken during KORUS-AQ at different times
of day (Figure 5). GEOS-Chem model values are in parentheses. The Table shows the AMF as computed from equation (3) and
the contributions from the diurnally varying factors affecting its computation at the three different times of the day including solar
zenith angle (SZA), relative azimuth angle (RAA), geometric AMF (AMF$_G$), and scattering correction factor ($\int_{0}^{z_T} w(z)S(z)\mathrm{d}z$, with
the scattering weights computed for clear sky). All computations use a surface albedo of 0.075, a viewing zenith angle (VZA) of
43.4°, and a surface pressure of 1013 hPa invariant with the time of the day.
[2] Local time (LT) is Korean Standard Time (KST). Solar noon was at 1220 KST during the KORUS-AQ period.



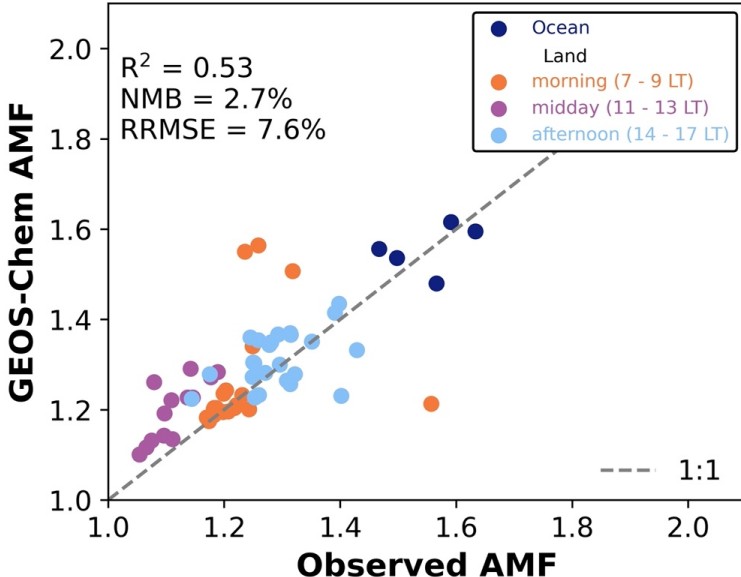

**Figure 6**. Variability in the air mass factor (AMF) for tropospheric NO$_2$ VCD retrievals from satellite as seen in 63 individual KORUS-AQ vertical profiles. The Figure compares GEOS-Chem and observed AMF calculated from equation (3). Coefficient of determination (R$^2$), normalized mean bias (NMB), and relative root-mean-square error (RRMSE) are given inset.