# Peer review of "Tropospheric NO2 vertical profiles over South Korea and their relation to oxidant chemistry: Implications for geostationary satellite retrievals and the observation of NO2 diurnal variation from space"

_EGUsphere, 2022_

## Author Comment (AC1)

Our response can be found in blue font.

**Reviewer 1**

Yang et al. used aircraft observations from the Korea-United States Air Quality (KORUS-AQ) campaign to get detailed insights into the diurnal variability and vertical distribution of tropospheric NO2. With this knowledge, Chemical Transport Model (CTM) simulations were optimized regarding mainly the oxidant chemistry. Furthermore, the authors analyzed the diurnal variability of the air mass factor (AMF), which is crucial when retrieving NO2 vertical column densities from the recently launched Geostationary Environment Monitoring Spectrometer (GEMS). This work is relevant since for the GEMS measurement campaign, a detailed knowledge about the AMF in this region is required and until know not known. The authors show, that their optimization of the CTM simulation within the GEOS-Chem model, significantly reduces the bias between observation and model. Nevertheless, especially in the planetary boundary layer (PBL), the model still does not reflect completely the observations. However, the AMF variability can be mostly captured by the GEOS-Chem model. There are only significant deviations in the morning due to incorrect timing of mixed layer growth. I recommend this paper to be published in Atmospheric Measurement Techniques, after the following minor points of criticism will have been addressed.

We thank the reviewer for the thoughtful and supportive comments. Our response to the general remarks and specific remarks is as follows:

**General remarks:**

It is hard to follow the description of the figures. Especially for Fig. 4 and 5, I would suggest a denotation with a/b/c/d and a clear reference when discussing trends and/or comparing data. The calculation of errors and their sources are in many cases not clear. Please clarify for the whole

The calculation of errors and their sources are in many cases not clear. Please clarify for the manuscript.

The revised text now includes denotation with a/b/c/d for Figures 1, 3, 4, and 5. The text and legends are updated accordingly to refer to the respective panels.

**Specific remarks:**

Line 103: ...which leads us to evaluation... --> ...which leads us to **the** evaluation... Thank you for catching this typo. Now the text is corrected.

Line 123: ..., A high-Resolution... --> a High-Resolution We fixed the revised manuscript as suggested.

Line 132: Introduce here the abbreviation of the version v13.3.4 and not in Line 145 The revised text reflects this suggestion.

Line 167: The sentence "The NO3- photolysis...x  $j_{HNO3}$ ." is not clear.

- EF relative to jHNO3 mostly means a ratio and not a multiplication
- please rephrase

In the revised text, it is now fixed as "They express the NO3- photolysis frequency  $(j_{NO_3})$  as the HNO3 photolysis frequency  $(j_{HNO_3})$  multiplied by an enhancement factor (EF):  $j_{NO_3} = EF \times j_{HNO_3}$ ."

Line 178: what does it mean "very high"? In comparison to what?

The revised text removed "very high" and changed to "Coarse particulate matter measured by the surface air quality network in South Korea is on average  $20 \ \mu g \ m^{-3}, \dots$ "

Line 205: Is this 50 % increase just an empirical estimation? Why this exact value? Yes, a 50% increase was an empirical estimation that helps to help with the underestimation issue. We simply took the ratio of CO values between observation and GEOS-Chem v13.3.4 to get 50%.

**Line 219: Is $z_t$ constant or date dependent?**

We assumed  $z_t$  to be constant. The following text is added to the revised text: "The tropopause is assumed to be at 12 km altitude, but results are insensitive to that assumption because the bulk of the NO2 tropospheric column is in the PBL."

**The index of $z_t$ must not to be in italics.**

Thanks for catching this typo. All  $z_t$  in the manuscript is corrected so that the index is no longer italicized.

**Line 259 and 261: How are the relative differences of the profiles calculated? At one given altitude? Or overall the PBL?**

The relative difference at each altitude bin was computed for altitude bin < 2km. Then the mean value of the relative difference at each altitude bin was computed. To clarify, we added a description of it in parentheses. "...was previously underestimated by 21% in the PBL (averaging over altitudes below 2 km) ..."

**Line 277: What is the accuracy of the measurement? Is it possible to show error bars or at least give an accuracy?**

Under section 2.1, we added the following sentence: "The stated uncertainties for these measurements are 5% for NO2, 30 pptv + 20% for NO, 5 ppbv + 10% for O3, 12% for  $j_{NO_2}$ , 32% for HOx, 0.8 s-1 for OHR, 2% (or 2 ppbv) for CO, 34% for sulfate, nitrate, and ammonium aerosols, and 30% for H2O2."

**Line 332ff: Is the described behavior only taken from literature (Chong/Crawford) or is it taken from the new data? Where is this time resolution shown? Figure 5 only shows the given 3 timeslots**

The described behavior in line 332ff is taken from literature only and not from the new data. To avoid confusion, we modified the sentence to be more explicit as follows: "Although surface NO2 peaks strongly in the morning (Figure 5 panel B), the NO2 VCDs observed from ground-based sun-staring column measurements in the SMA from Chong et al. (2018) and Crawford et al. (2021) show an increase from early morning to 11 LT, steady concentrations until 14 LT, and a slight decrease afterward."

**Line 353: mention, that the model values are given in parentheses in Table 1.**

We fixed the text as suggested.

**Line 373f: please rephrase the sentence.**

The sentence is rephrased as follows: "The AMF RRMSE from using GEOS-Chem model NO2 shape factors (Figure 6) can be placed in the context of other contributions to the AMF error."

**Line 364f: what are the values given in parentheses? Mean+StdDev?**

Yes, we changed the revised text to make this clearer: "Oceanic profiles have higher AMF (Mean: 1.55, SD: 0.068) than land (Mean: 1.23, SD: 0.097) ...".

**Line 372: Which errors do you mean? The RRMSE from above?**

Yes, the errors mean RRMSE in the context. The revised text now reads as "The AMF RRMSE from using GEOS-Chem model  $NO_2$  shape factors, as shown in Figure 6, can be placed in the context of other contributions to the AMF error."

**Reviewer 2**

This manuscript investigates the effect of the diurnal variability of NO2 vertical structure on the air mass factor. It is a nice albeit brief investigation into this topic. I recommend publication after minor revisions.

The lone major suggestion is to add-on to Figure 5 to better show the diurnal variability of NO2 at various altitudes. It's hard to tell whether GEOS-Chem is getting the diurnal patterns correctly. I do think this is a critical missing piece of the manuscript because it would more clearly demonstrate whether NO2 vertical mixing in GEOS-Chem needs to be further improved (in future work). I expand on this further in my comments below.

The Introduction also needs several minor adjustments, but otherwise the manuscript is in good shape.

We thank the reviewer for helpful and supportive comments. Our response to the specific comments is as follows:

Line 22. Modify "high-density" to different phrase. Maybe "hourly high-spatial resolution" Thank you for suggesting a better phrase. The sentence is modified and no longer includes the term "high-density".

Line 22. I recommend merging sentence #2 of this paragraph into the current sentence #4 of this paragraph. The current sentence #2 is a bit out of place, and should belong in a sentence mentioning the study motivation. Similar to the sentence in Line 50.

We revised sentence #2 to be more motivational. Now the sentence reads as follows: "The Geostationary Environment Monitoring Spectrometer (GEMS) is now providing hourly  $NO_2$  satellite observations over East Asia, offering the first direct measurements of  $NO_2$  diurnal variation from space to guide understanding of  $NO_x$  emissions and chemistry." Line 50 is also revised accordingly.

Line 32. Mention "SMA" somewhere in this sentence.

Fixed the sentence to include SMA as suggested.

Line 41. Never heard of NOx referred to as radicals, but I suppose an argument could be made. Instead I recommend modifying "Nitrogen oxide radicals" to "Nitrogen oxides" Now it is fixed as nitrogen oxides. Line 48. Modify "starting to provide" to "providing" Thanks for the suggestion. It's changed as suggested.

Line 61. Modify "GEMS is the first geostationary instrument" to "GEMS is the first geostationary instrument measuring trace gases, such as NO2."

Reviewer 3 gave a similar suggestion to make this sentence clearer. Now the text reads as "GEMS is the first geostationary instrument to measure tropospheric  $NO_2$ ."

Line 62. Modify "from which the stratospheric portion is removed" to "from which the stratospheric portion is removed to estimate the tropospheric SCD" The revised text now reflects the suggestion.

Line 69. Modify "observed" to "observe" Thanks for catching this typo. It is now fixed.

Line 78 - 80. Remove the sentence "These diurnal variations…". It is confusing and out of context in my opinion. Penn and Holloway investigated a less polluted area in the US, and should not be compared to Seoul in this context. Subsequently modify "this issue" in Line 81 "the diurnal patterns of NO2"

The revised text now reflects the suggestion.

Line 205. It would be appropriate to add one more sentence here describing what the previous literature found. Are global CO emissions too low? Is the lifetime of CO too short? Or is the reason for the underestimate still not known?

We modified the text to include more details about CO underestimation. Now the text reads as follows: "This could be due to missing CO sources in East Asia (Gaubert et al., 2020; Park et al., 2021) missing VOC emissions (Huijnen et al., 2010), and/or excessive OH concentrations (Naik et al., 2013). Here we find that a 50% increase at all altitudes in the GEOS-Chem CO concentrations used as boundary conditions corrects the model bias for simulating KORUS-AQ observations. We adopt this correction in our simulation."

Figure 5 (or a new figure). It would be very interesting to create a pseudo diurnal plot from this information. For example on the x axis would be time of day, and y-axis would be number density. Both the observations and GEOS-Chem would be plotted on the same panel. And then have 4 or 6 panels with the diurnal patterns at 4 or 6 different altitudes (surface+three different heights) or surface+five different heights). Currently I cannot tell how well GEOS-Chem is reproducing the diurnal pattern at each altitude (only the absolute values).

We tried to do this but we only have three points to characterize the diurnal variation and so that would be a lot of panels with sparse data. We also tried a contour plot with the time of day as the x-axis and altitude as the y-axis, but it didn't work for the same reason. We switched the quantity plotted from number density to shape factor to focus more on what we care about for the AMF calculation.

Line 348. To complete this paragraph, it'd be best to mention how this will affect the VCD in a final sentence. Since VCD and AMF are inversely correlated, a 14% AMF increase will decrease

the VCD 14% more in the afternoon than at mid-day, assuming an identical slant column at both hours.

We prefer not to act on this suggestion. VCD and AMF are not necessarily inversely correlated – VCD depends on the absolute amount and AMF depends on the shape. There is no reason to assume an identical slant column at different times of the day.

Line 384. Modify "it would be rare to find a 7×8 km2 GEMS pixel without clouds" to "it would be likely that a 7×8 km2 GEMS pixel would have some amount of clouds" This sentence is removed from the text.

**Reviewer 3**

In this manuscript, the authors report on a study comparing air mass factors for GEMS tropospheric NO2 satellite retrievals over Korea based on measured and modelled NO2 profiles. The measurements are 63 vertical NO2 profiles from the KORUS-AQ campaign, while the modelled profiles are from a dedicated version of the GEOS-Chem model. A particular emphasis is on the evaluation of diurnal changes in the vertical NO2 profiles and thus the air mass factors. The main conclusions from the paper are that the NO2 AMFs over Seoul significantly increase over the course of the day and that the new version of the GEOS-Chem model predicts NO2 profiles, which lead to AMFs, which are in good agreement with those based on observed profiles.

The topic of the manuscript is relevant for the atmospheric community, as the diurnal variability of the AMF needs to be understood and modelled well in order to make use of the exciting new observations from geostationary orbit, of which the Korean GEMS instrument is the first to be in operation. The manuscript is overall well written and contains clear figures illustrating the main findings. There is, however, a number of points, which the authors need to address before the manuscript can be accepted for publication.

We thank the reviewer for the thorough review of the manuscript and helpful comments. Our response to general comments and detailed comments can be found below:

**General Comments**

Throughout the manuscript, the wording is in my opinion not as accurate as it should be. There are many occasions where it is said that the "GEOS-Chem calculated AMFs are …" while I think the formulation should be "AMFs based on GEOS-Chem profiles …". Some more examples from the abstract and summary:

For the last sentence of the Introduction, we changed the sentence to the suggested phrase. "We go on to examine the diurnal variation of NO2 vertical profiles and the resulting diurnal variation of AMF as defined by the observations and by GEOS-Chem, and from there quantify the AMF errors associated with retrieving NO2 diurnal variations from GEMS using GEOS-Chem vertical profiles."

"the ability of the GEOS-Chem CTM to compute the AMF" – GEOS-Chem computes the NO2 profiles, not the AMFs.

The text is now fixed as follows: "Here, we use aircraft observations from the Korea-United States Air Quality (KORUS-AQ) campaign over the Seoul Metropolitan Area (SMA) and around the Korean peninsula to better understand the factors controlling the NO2 vertical profile, its diurnal variation, the implications for the AMF, and the ability of the GEOS-Chem CTM to compute the NO2 vertical profiles used for AMF."

"The KORUS-AQ vertical profiles indicate that 95% of the tropospheric NO2 VCD detected from space over the Seoul Metropolitan Area (SMA) originates from the PBL" – The VCD (if fit is correct) does not depend on how it is detected. What is meant here is probably what the authors earlier called "cumulative SCD".

Thank you for pointing out this typo. We corrected the sentence as follows: "The KORUS-AQ vertical profiles indicate that 95% of the cumulative slant column density (SCD) detected from space over the Seoul Metropolitan Area (SMA) originates from the PBL, despite the much higher sensitivity to NO2 in the free troposphere, reflecting the highly polluted conditions."

"This leads to a large diurnal variability of NO2 detected from space" – first of all, I'm not sure if a variation of 20% is large for NO2, and second, it is not clear how this is the result of most of the NO2 being located in the PBL.

We agree that there's no link in the degree of diurnal variability and 95% of the cumulative SCD originates from the PBL. Hence, we modified the sentence as follows: "The diurnal variation of NO2 VCD observed by GEMS should thus provide direct insights on NOx emissions and PBL chemistry in urban areas of East Asia, but this requires correct accounting of the diurnal variation in AMF."

I think all authors should read the text carefully again and make sure that the wording is precise.

Thanks for your suggestion. We modified the wording precisely in this round of revision.

One of the main points of the paper is the improved GEOS-Chem version, which supposedly provides better NO2 profiles for AMF calculations of Asia. However, while the agreement of the new model version with some observed trace gases is improved, this is not obvious for NO2. I therefore suggest to add the AMFs calculated by the standard version of GEOS-Chem in Figure 4 and Table 1 to demonstrate the improvement if there is any or else explicitly discuss the lack in improvement.

The main point of improving the GEOS-Chem version for the first half of the paper is to make sure that we simulate the NO2 vertical profile well for the right reason (i.e., well-represented oxidant chemistry). In the standard model, the oxidant chemistry in South Korea is not well-represented with underestimated O3 and HO2 (Figure 1). However, once we modify the GEOS-Chem model, the oxidant chemistry is better represented (i.e., O3 and HO2 simulations improved). Hence, we can be confident that oxidant chemistry driving NO2 better represents South Korea's environment now. To better convey this to the readers, we added the following sentence towards the end of the Introduction: "As we will see, these updates enable a much-improved representation of oxidant chemistry relative to previous GEOS-Chem versions, which provides a basis for model evaluation with observed NO2 concentrations and NO/NO2 ratios."

For Figure 4, we added standard model AMF values as well as the line in the plot as the reviewer suggested. However, the differences we see in standard model vs. modified model profiles and hence, AMF computed based on those profiles are minor. The text is modified as follows: "Figure 4 shows the median vertical profile of simulated (red line), observed (black line), and standard

GEOS-Chem v13.3.4 (blue line) NO2 number density in the SMA during KORUS-AQ (panel B), along with the scattering weights ... The differences between our updated GEOS-Chem simulation and the standard GEOS-Chem v13.3.4 are minuscule." For Table 1, the scattering correction factor based on modified GEOS-Chem profiles as well as AMF based on modified GEOS-Chem profiles are in very close agreement with the observation. Hence, we didn't include the AMF based on the standard model in Table 1.

Throughout the manuscript it is emphasised, how the new version of the model performs better over Asia, and that GEOS-Chem profiles result in AMFs, which are very close to those derived from measured profiles. While I do not question this, I think it should be stated very clearly, that the KORUS-AQ profiles were all taken during one season and in a very limited number of places, and it is not at all clear if the nice results shown here can be reproduced for another location in another season. To some degree, this model version has been tuned for KORUS-AQ, and I expect problems when it is applied under different conditions.

In the Conclusions section, we added the following sentence to address this comment: "Further evaluation of the GEOS-Chem NO2 profiles over East Asia should be conducted with aircraft campaigns for other regions and seasons, and opportunities will arise with the ASIA-AQ campaign over Korea in the winter of 2024 (NASA, 2021)."

Somewhere in the manuscript, it should be stated, that all AMF calculations were made for a Rayleigh atmosphere without aerosols (at least I assume that this is the case).

We modified the last paragraph of section 2.3 as follows: "We focus our attention on clear-sky scenes for a Rayleigh atmosphere without aerosol scattering."

Detailed comments

Line 61: "GEMS is the first geostationary instrument" => GEMS is the first geostationary instrument to measure tropospheric NO2 The revised text now reads as suggested.

Line 70: "tropospheric NO2 VCDs vary with the time of day as driven by .. mixing depth" => I do not think that mixing depth is a driver for the VCD

Thanks for catching this typo. The text now removed the mixing depth.

Line 187: "we find in GEOS-Chem ..:" Isn't that a matter of the emission inventory, and not the CTM?

Yes, it's a matter of emission inventory in GEOS-Chem such as HEMCO in our case.

Line 352: "GEOS-Chem reproduces closely the observed diurnal variation of the scattering correction factor" => GEOS-Chem reproduces the profile, not the scattering factor

Now the revised text reads as follows: "We see from Table 1 that GEOS-Chem (given in parentheses) reproduces closely the observed diurnal variation of the shape factor and thus of the scattering correction factor and the AMF."

Line 363: "surface type" => while this is formally OK, the driver for the difference is the NO2 profile shape, determined by the presence or absence of local sources.

Now the revised text reads as follows: "The two main drivers of this observed AMF variability are  $NO_2$  profile shape, determined by the presence or absence of local sources (ocean versus land), and time of day for land."

Line 393: "they are not an obvious source of error when comparing model and observed NO2 profiles" => this probably depends on whether the scene in GEOS-Chem is cloudy or not The text is now revised as follows: "Although clouds would be expected to affect the NO2 vertical profile through photolysis, chemistry, and vertical motions, these effects are complex, and we do not find obvious cloud-driven differences in model or observed NO2 profiles."

Line 394: "the afternoon formation of fair-weather cumuli would decrease the sensitivity of the satellite measurement to the PBL and therefore alias the observed diurnal variation of NO2." => yes, but only if no cloud correction is applied in the retrieval.

We added a clarification to this sentence as follows: "From a diurnal variability perspective, the afternoon formation of fair-weather cumuli would decrease the sensitivity of the satellite measurement to the PBL (which would not be detected in the cloudy fraction of the scene) and therefore alias the observed diurnal variation of NO2."

Line 401: "to better understand how the vertical distribution of NO2 affects the air mass factors (AMFs) for satellite retrievals of tropospheric NO2 vertical column density" => I think that the effects of NO2 vertical distribution on AMFs are well understood. Isn't the point here to look at the effect of diurnal changes in the vertical NO2 distribution on the variation of AMFs over the day?

Thank you for pointing this out. Yes, this is exactly the purpose of the work. We modified the sentence to the following: "We used extensive observations of NO2 vertical profiles and related chemistry from the KORUS-AQ aircraft campaign over and around South Korea in May-June 2016 to better understand how the diurnal changes in the vertical distribution of NO2 affect the diurnal variation of air mass factors (AMFs) used for satellite retrievals of tropospheric NO2 vertical column density (VCD) in the polluted East Asia chemical environment."

Figure 1: I find it odd to mix volume mixing ratios and mass concentrations in one figure. The unit of Figure 1 is now  $\mu g/m^3$  for both the x-axis and the y-axis.

Figure 2: While the median profiles are nice, it would be good to add a figure also indicating the variability of these profiles

We tried adding horizontal bars to show variability, but it made a mess of the Figure.

Figures 2 - 5: There are some strange steps in these curves – what is the vertical resolution / sampling?

The vertical resolution is 0.25 km for  $z \le 2$ km and 1 km for z > 2km. We tried to make the vertical resolution to be finer than 0.25 km within PBL, but it was not feasible as observation points decreased significantly and we had to make interpolations when the vertical resolution became finer. We added the following description to Figure 2 caption: "Here and in the following Figures, vertical profiles are constructed by binning the data in 0.25-km vertical intervals below 2 km altitude and 1-km vertical intervals above 2 km altitude."